# Single Crystal FLIM Characterization of Clofazimine Loaded in Silica-Based Mesoporous Materials and Zeolites

**DOI:** 10.3390/ijms20122859

**Published:** 2019-06-12

**Authors:** Lorenzo Angiolini, Boiko Cohen, Abderrazzak Douhal

**Affiliations:** Departamento de Química Física, Facultad de Ciencias Ambientales y Bioquímica, and INAMOL, Universidad de Castilla-La Mancha, Avenida Carlos III, S/N, 45071 Toledo, Spain; Lorenzo.Angiolini@uclm.es

**Keywords:** MCM-41, SBA-15, zeolites, mesoporous materials, silica-based materials, FLIM, microscopy, drug delivery, H- and J-aggregates

## Abstract

Clofazimine (CLZ) is an effective antibiotic used against a wide spectrum of Gram-positive bacteria and leprosy. One of its main drawbacks is its poor solubility in water. Silica based materials are used as drug delivery carriers that can increase the solubility of different hydrophobic drugs. Here, we studied how the properties of the silica framework of the mesoporous materials SBA-15, MCM-41, Al-MCM-41, and zeolites NaX, NaY, and HY affect the loading, stability, and distribution of encapsulated CLZ. Time-correlated single-photon counting (TCSPC) and fluorescence lifetime imaging microscopy (FLIM) experiments show the presence of neutral and protonated CLZ (1.3–3.8 ns) and weakly interacting aggregates (0.4–0.9 ns), along with H- and J-type aggregates (<0.1 ns). For the mesoporous and HY zeolite composites, the relative contribution to the overall emission spectra from H-type aggregates is low (<10%), while for the J-type aggregates it becomes higher (~30%). For NaX and NaY the former increased whereas the latter decreased. Although the CLZ@mesoporous composites show higher loading compared to the CLZ@zeolites ones, the behavior of CLZ is not uniform and its dynamics are more heterogeneous across different single mesoporous particles. These results may have implication in the design of silica-based drug carriers for better loading and release mechanisms of hydrophobic drugs.

## 1. Introduction

The increase in resistance to antibiotics presents new challenges to fight bacterial infections [1,2,3]. A lot of efforts have been directed at developing administration methods of drugs with diminished efficiency due to their low solubility and degradation processes in the gastrointestinal tract. The use of nanocarrier-based drug delivery systems improves the solubility and bioavailability of many drugs [4]. In this way, these systems are capable of releasing the required amount of antibiotic while maintaining its effectiveness, thus helping to reduce the risk of developing resistance.

Clofazimine (CLZ) is an antibiotic used mainly to treat leprosy, but it is also known to be effective against a broad spectrum of Gram-positive bacteria and multidrug resistant cancers [5,6]. However, CLZ efficiency is affected by poor solubility in water, which prevents its intravenous administration and it is not well absorbed by the gastrointestinal tract, which leads to its accumulation into body tissues and skin coloring [7,8]. Several reports have shown that CLZ solubility in aqueous solutions increases when encapsulated or complexed with nanocarriers, resulting in a higher bioavailability and antibacterial efficiency [5,9,10,11,12]. The nanoconfinement effect exerted by mesoporous silica materials can stabilize the loaded drug in its amorphous state [13], which grants greater molecular motion and enhanced thermodynamic properties, compared to its crystalline state, and provides higher apparent solubility and dissolution rate [14]. However, for the delivery of CLZ, it is necessary to develop carriers able not only to increase its solubility, but also, most importantly, capable of delivering the antibiotic to the stomach and intestines without early release.

Silica-based mesoporous materials, such as mesoporous silica particles (MSPs), MCM-41, and zeolites, have been used to encapsulate and deliver drugs [15,16,17,18,19,20,21,22]. These materials present high pore volume, large surface area, and tuneable pore diameter that is translated to high loading capacity. Additionally, they are chemically and thermally stable, making them suitable materials for drug delivery applications. The MSPs have been successfully used to solubilize CLZ and improve its delivery in simulated gastric fluids, thus obtaining effective antimicrobial concentrations in infected macrophages [14,23,24]. Additional studies have shown that the MSPs can be biocompatible vessels for drug delivery, with stable and durable encapsulation of the drugs [25,26,27,28]. In a recent study, we reported on how the surface properties of MSPs (hydrophobic and hydrophilic, with 1% and 5% *w*/*w* of silanol groups, respectively) affect the loading, distribution, and release of CLZ in water at different pHs [23]. We observed that the amount of silanol groups modifies the type and strength of molecular interactions (CLZ–CLZ and CLZ–silica framework), causing a higher loading %, higher affinity for aggregate formation, and slower release within the hydrophobic MSPs, where the amount of silanol groups is lower.

Here, we report on the interaction of CLZ with various amorphous mesoporous silica materials, MCM-41, Al-MCM-41, SBA-15, and crystalline zeolites NaX, NaY, and HY, observing how different surface properties and crystallinity affect the CLZ loading and conformation within the channels of these materials (Scheme 1). We performed steady-state absorption and emission experiments on the loaded materials in dichloromethane (DCM) suspensions, along with a time-correlated single-photon counting (TCSPC) decay analysis. Additionally, we collected fluorescence lifetime images (FLIM) and measured the emission spectra and decays of single particles of solid hybrid materials. We observed stronger interactions with MCM-41, Al-MCM-41, and SBA-15 rather than with the zeolites, because of the higher amount of silanol and OH groups that can freely interact with CLZ. The interaction results in the protonation of CLZ. This process is observed also when CLZ is encapsulated in the channels of the NaX and NaY zeolites, but the presence of Na^+^ prevents stronger interactions with the silica framework. The lower loading (~50%) is a consequence of the smaller pore size and the Na^+^ screening effect. Thus, the nature of the silica-based host materials can affect the conformation of CLZ and alter its properties, modifying its loading and release efficiencies.

## 2. Results and Discussion

### 2.1. Steady-State Absorption and Diffuse Transmittance Spectra

To begin with, we measured the UV-visible absorption spectra of CLZ in a DCM solution (4.3 × 10^−5^ M) along with the diffuse transmittance spectra (DTS) of the CLZ@silica-based materials in DCM suspensions (Figure 1 and Appendix A, panel A). The absorption spectrum of CLZ in DCM is characterized by a band at 449 nm and a second one at 520 nm. The change of the concentration to 4.8 × 10^−6^ M and 4.3 × 10^−7^ M did not affect the absorption spectrum. The DTS of the composites have the main band red shifted in comparison with CLZ in DCM solution. For MCM-41, it is located at 504 nm, shifted by 2430 cm^−1^, for CLZ@Al-MCM-41 and CLZ@SBA-15 at 502 nm (shifted by 2351 cm^−1^), for CLZ@NaX and CLZ@NaY at 490 nm (shifted by 1864 cm^−1^), while for CLZ@HY it is located at 516 nm, which corresponds to a red shift of 2892 cm^−1^. The DTS spectra show also the presence of a band at 385 nm, independently of the type of the silica-based support. Finally, for NaX and NaY composites, an additional band at 470 nm is observed along with a red edge tail at wavelengths longer than 700 nm. 

We assign the band centred at 449 nm of CLZ in DCM to the neutral form of CLZ, as reported also for CLZ in water at pH 7 and in DMSO [24,29,30]. The red shift of the main absorption maximum along with the presence of the second band/shoulder at 385 nm indicates that upon interaction with the silica materials CLZ is mainly present in its protonated form. In water, after protonation (pH 4), CLZ presents two bands centred at 495 nm and 373 nm (less intense). Additional decrease in pH causes a second protonation and the maximum of absorption shifts to 540 nm [30]. For the MCM-41, Al-MCM-41, and SBA-15 composites, the red shift of the maxima of absorption is 2351–2430 cm^−1^, compared to CLZ in water at pH 4 (2070 cm^−1^). The larger shift can be explained by the presence of nonspecific interactions with the more hydrophobic environment of the pores and/or by the specific interactions with the OH and silanol groups present on the surface [22,31]. Notably, FTIR measurements to analyze the contribution of silanol groups to the interaction of the silica materials with CLZ did not provide reliable information due to the low concentration (4% *w*/*w*) used for the studies in this work (Appendix A). Several studies have reported that CLZ can form intermolecular H-bonds with materials having proton donor and/or acceptor characteristics [32,33,34]. The high number of OH and silanol groups in SBA-15, MCM-41, and Al-MCM-41 can provide sites for intermolecular H-bonding interactions, contributing to the observed red shift of the CLZ^+^ absorption maxima. 

The smaller diameter of the NaX and NaY zeolites pores (13 Å) interconnected by channels of 8 Å [35], compared to MCM-41 (25 Å), Al-MCM-41 (25 Å) and SBA-15 (60 Å) pores (Scheme 1), can result in only partial encapsulation of CLZ (7.5–9.2 Å) and thus prevent the protonation of the imino group. This in turn can explain the reduced intensity of the band at 385 nm and the presence of the shoulder at 470 nm, which can be related to the neutral form of CLZ. Additionally, in NaX and NaY, the Na^+^ counterion can screen the protonated CLZ^+^ within the pores [36,37], reducing its interaction with the zeolites framework or directly destabilize it through electrostatic interaction and causing the weaker red shift of the CLZ maxima (490 nm). 

Finally, in the HY zeolite, the counterion is H^+^ since heating over 400 °C causes NH_3_ to evaporate, leaving a positive charge that is injected in the silica framework. As a result, this zeolite is characterized by a Brønsted acidity, whereas the Na^+^ counterion results in a Lewis acidity of the silica framework [38]. The different size of the counterions reduces the screening effect observed in NaX and NaY, allowing better interaction and stabilization of CLZ within HY channels, while the change in the acidity from Lewis to Brønsted contributes to the additional red shift of the main DTS band (516 nm). Aggregation can also affect the CLZ behavior in solution and in confined spaces [7,23,30]. J-type aggregates usually contribute to the red shift of the maximum of absorption upon stabilization of the involved species, whereas H-type aggregates give rise to a new band at shorter wavelengths [39]. As we observe the band at 385 nm and an increasing contribution at the red side of the spectra, we must consider that aggregate populations can also contribute to the absorption spectrum of the encapsulated CLZ.

### 2.2. CLZ Loading

The smaller pore sizes of the zeolites and the presence of a counterion screening effect in their channels also reduce the loading capacity of these materials compared to SBA-15, MCM-41, and Al-MCM-41 (Table 1). In fact, NaX presents small pore size (13 Å) and the highest amount of Na^+^ [40,41], which leads to only 41% of CLZ to be loaded, while NaY (13 Å) has a reduced Na^+^ amount, which allows up to 51% of CLZ to be loaded. For HY, the combination of the H^+^ screening effect and the smallest pore size (7.4 Å) only allows for 46% of CLZ to be loaded. As the size of CLZ (7.5–9.2 Å) is comparable to the size of the pores (7.4–13 Å) of the zeolites frameworks and the windows interconnecting them (8 Å), a high deposition of CLZ on the outer surface of these materials can be expected. However, following four rinsing cycles with DCM to remove the loose CLZ molecules, we expect that the remaining ones interact with the pores and are the ones mostly contributing to the photobehavior of the composites.

On the other hand, the amorphous composites of SBA-15, MCM-41, and Al-MCM-41 present a significantly higher loading of CLZ (>99%, 84% and 81%, respectively). These materials have larger pore size and present a neutral framework where the Na^+^/H^+^ screening effect is not observed, thus resulting in a significantly higher loading capacity.

Considering the total pore volume of the materials (Table 1), the amount of antibiotic loaded in SBA-15 is higher than in MCM-41 and Al-MCM-41. In SBA-15, despite the reduced volume (0.7–0.9 cm^3^ g^−1^) compared to MCM-41 and Al-MCM-41 (1 cm^3^ g^−1^), the loading was higher. This could be related to the presence of structural defects on the surface of the pores of SBA-15 that provide a microporous environment with higher apparent silanol density [45], which can strengthen the interactions with CLZ, thus increasing loading. The pore volumes of the zeolites are comparable (0.29, 0.34, and 0.33 for NaX, NaY, and HY, respectively) and further demonstrate that the pore size and the counterion screening effect are the properties affecting the loading of CLZ.

### 2.3. Steady-State Emission Spectra

We could not record reliable emission spectra of CLZ in DCM solution due to the very weak intensity, which renders it impossible to differentiate between the CLZ signal and the instrumental noise. Figure 1a and Appendix A show the steady-state emission spectra of CLZ@silica material suspensions in DCM upon excitation at 470 nm. The CLZ@MCM-41 and the CLZ@Al-MCM-41 composites both show a band with maximum of intensity at 525 nm and a second, less intense one at 555 nm and 560 nm, respectively. The maximum at 560 nm is more intense for CLZ@Al-MCM-41. The CLZ@SBA-15 composite shows only a broad band with a single maximum at 570 nm. The CLZ@NaX zeolite presents a maximum at 530 nm and a less intense maximum at 560 nm, while the CLZ@NaY zeolite shows the maximum of intensity at 560 nm and a less intense maximum at 530 nm. Finally, the HY zeolite presents maximum intensity at 570 nm and a less intense shoulder at 540 nm.

Several studies have reported on the emission spectrum of CLZ in DMSO, which presents a broad band with two maxima at 540 and 580 nm [29,46]. The lack of CLZ emission in other solvents can be related to fluorescence quenching due to formation of dimers and aggregates. The formation of these dimers and/or aggregates is driven by intermolecular interactions such as π-stacking and weak H-bonds (C-H^…^N, C–H^…^π, and C–H^…^Cl interactions) [6,32,47,48]. The H-bonding properties of DMSO can disrupt the CLZ–CLZ H-bond or noncovalent interactions [49]. Instead, tetrahydrofuran (THF) is reported to affect the bonding ability of CLZ molecules to form CLZ-polymorphs through the electron donating ether group, which can interact with CLZ chlorine-substituted benzenes and cause solvation of the molecules [6]. However, no emission spectra have been recorded in THF. Since DCM cannot significantly alter the CLZ–CLZ interactions lacking the H-bonding properties of DMSO or the electron donating ability of THF, the emission in this solvent is considerably quenched. On the other hand, the suspensions of the studied CLZ composites show stronger emission, which is a result of the encapsulation of the CLZ molecules that reduces the possibility of formation of aggregates.

The emission spectra of the loaded composites show broad bands with multiple emission maxima, suggesting the presence of several CLZ species in the channels of the silica-based materials and can be related to the acidity of the hosts. For MCM-41, which presents Lewis acidity, the emission spectrum of the composite shows a maximum intensity at 525 nm and a broad shoulder at longer wavelengths. For CLZ@Al-MCM-41, where the Al substitution increases the acidity providing Brønsted acidic sites, we observed an increased emission intensity at the red side of the spectra. The increase of the emission intensity at the red side of the spectrum becomes more evident for the zeolites where the lower Si/Al ratio (<2.47) provides additional Brønsted acidic sites compared to the mesoporous materials. However, for NaX and NaY, the presence of Na^+^ as the counterion tends to promote Lewis acidic sites, as opposed to HY where the Brønsted character is more evident. As consequence, the CLZ@NaX composites are characterized by emission maximum below 550 nm but the red edge emission becomes more defined and the band broadens. For the CLZ@NaY composite, the intensity of the band at 525 nm further decreases, while the one at the less energetic wavelength shifts to 560 nm and becomes more intense. Finally, for CLZ@HY, the red edge band is further red-shifted to 580 nm, while the more energetic one has the lowest intensity. For CLZ@SBA-15, we observed a broad band with single emission maximum at 570 nm. The lack of a well resolved emission band can be a consequence of aggregation due to the highest loading (>99%) or due to the increased freedom for CLZ species (monomeric and aggregates) to interact with each other in the large SBA-15 pores (60 Å). Alternatively, the SBA-15 surface, compared to MCM-41, is characterized by considerable roughness due to the presence of structural defects that increase the apparent silanol density [45], thus further increasing the heterogeneity of CLZ interactions with the silica framework.

By comparing this behavior with the reported emission spectra of CLZ in DMSO, we assign the high energy emission below 550 nm to the neutral form of the encapsulated drug, while the emission at the red side of the spectrum corresponds to its protonated form. It should be noted that the emission spectra for the studied composites are broad, which is associated with the heterogeneous distribution of the encapsulated guest giving rise to different specific and nonspecific interactions between the CLZ molecules and/or with the support. Additionally, we cannot exclude the presence of different types of aggregates (H- and/or J-aggregates) as a result of the limited space provided by the host channels.

### 2.4. TCSPC Decays in DCM Suspensions

To obtain information on the dynamics of the CLZ species encapsulated in the silica-based materials in DCM suspensions, we performed picosecond (ps) measurements upon excitation at 430 nm. 

Figure 1b shows the emission decays of the studied composites collected at 550 nm, while Table 2 gives the values of the time constants, and their relative pre-exponential factors obtained from a multiexponential fit of the experimental data collected at different observation wavelengths. All the composites show bi-exponential behavior with τ_1_ = 0.1–0.4 ns and τ_2_ = 0.9–2.2 ns. The value of τ_1_ is almost independent of the host type (with the notable difference being HY, where it becomes 0.4 ns). On the other hand, the values of τ_2_ are more sensitive to the type of the silica-based material. While in MCM-41 and Al-MCM-41 its value is 0.9–1.1 ns, it becomes 2.0–2.2 ns in NaX, NaY, and HY. For the CLZ@SBA-15 composite, the value of τ_2_ is intermediate (1.4–1.8 ns). Similar behavior has been observed for CLZ loaded in MSPs [23]. Additionally, a bi-exponential fluorescence decay for CLZ in DMSO solution has been reported with lifetimes of 0.11 ns and 2.35 ns [29]. 

We assign the short time component τ_1_ = 0.1–0.4 ns to dimeric/aggregate species and the longer component, τ_2_ = 0.9–2.2 ns, to monomeric CLZ encapsulated in the channels of the silica-based materials. Although the values of τ_1_ are comparable for all the samples, its relative contribution differs across the composites. For CLZ@SBA-15, the contribution of the aggregates is over 90%, which is in agreement with the presence of large pores that facilitate the formation of aggregation and strong CLZ–CLZ interactions. The observed behavior can be related also to the high loading efficiency of SBA-15 (>99%), which would favor the CLZ–CLZ interactions. The contribution of τ_1_ for CLZ@MCM-41 and CLZ@Al-MCM-41 is below 60%, suggesting that the population of aggregates is lower because of a stronger interaction of CLZ with the framework of these materials. The decrease in the value of τ_2_ for the CLZ@MCM-41 and CLZ@Al-MCM-41 composites (0.9 ad 1.1 ns, respectively) in comparison to the one for CLZ@SBA-15 (1.4–1.8 ns) is further evidence for the stronger interaction between CLZ and the Al-/MCM-41 hosts framework. The smaller pores of MCM-41 and Al-MCM-41 and the lack of defects and folding on the surface, compared to SBA-15, can reduce the formation of CLZ aggregates providing less aggregation sites. 

For the CLZ@NaX composites, the relative contribution of τ_1_ is 94% at 525 nm and it decreases to ~60% at longer wavelengths of observation. As suggested by the emission spectra, the neutral species of CLZ emit at the blue side of the spectrum (below 550 nm), while the protonated CLZ is contributing at the red side. The presence of significant amount of Na^+^ in the channels of NaX does not affect the neutral form of CLZ, favoring aggregate formation (a_1_ = 94% at 525 nm), while it can reduce the intermolecular interactions between the protonated CLZ molecules, resulting in the observed decrease in the relative contribution at 550 nm and higher. Similar behavior is observed for the CLZ@NaY composites. The contribution of τ_1_ for CLZ@NaY is high at 525 nm where the neutral form emits (a_1_ = 94%), and it decreases to 85% at longer wavelengths. The decrease is smaller compared to CLZ@NaX due to the lower concentration of Na^+^ in the NaY channels. For both zeolites, the presence of Na^+^ results in a weaker interaction with the host framework, similar to the CLZ@SBA-15 composites, thus leading to higher contribution from aggregates and longer lifetimes for the weakly interacting monomers (τ_2_ = 1.7–2.0 ns). Finally, the CLZ@HY composites show the highest values for both τ_1_ (0.2–0.4 ns) and τ_2_ (2.2 ns), with comparable relative contributions. The higher value of τ_2_, assigned to the CLZ monomers, suggests that the coupling between the guest molecules and the HY host framework is weaker, while the lower relative contribution of τ_1_ (~50%) in comparison with the rest of the studied composites is indicative of lower aggregation efficiency. This behavior can be explained in terms of the smaller channel size (7.4 Å), which can result in only partial encapsulation of the CLZ guest, thus affecting its tendency to form aggregates.

### 2.5. Solid State Scanning Confocal Fluorescence Microscopy

To better understand the behavior of CLZ encapsulated in the channels of the studied materials, we performed scanning confocal fluorescence microscopy, collecting the fluorescence lifetime images (FLIM) of the loaded composites, along with their emission spectra and TCPCS decays (Figure 2, Figure 3, Figure 4 and Figure 5 and Appendix A). The FLIM images of the studied composites with sizes of ~2 µm or smaller following excitation at 470 nm show a homogenous distribution of the emission intensity. This indicates that the loaded CLZ occupies all the host volume and is homogenously attached to the framework of the materials. 

#### 2.5.1. Single Particle Emission Spectra

Figure 2, Figure 3, Figure 4 and Figure 5 (panel B) and Appendix A (panel B) show representative emission spectra of the solid composites loaded with CLZ, upon excitation at 470 nm, while the single spectra for each material are reported in Appendix A. The averaged spectra of SBA-15, MCM-41, and Al-MCM-41 (Appendix A) present two bands with maxima at 595 nm (full width at half maximum, FWHM ~ 2000 cm^−1^) and 700 nm (FWHM ~ 950 cm^−1^), respectively. On the other hand, the zeolites present a broad band with maximum of intensity at 570 nm (FWHM ~ 2500 cm^−1^), and additional less intense maxima at 605 nm and 700 nm. In a recent study on the interaction of CLZ with mesoporous silica particles, we reported on the dependence of the emission spectra on the CLZ loading. In this report, we assigned the band at 570–600 nm, which was the dominant one in the spectra at lower CLZ loading, to monomer-like species, while the band at 700 nm, which presented the stronger intensity at higher CLZ loading, was assigned to aggregate-like species [23]. Further analysis of the intensity of the band maxima shows the different behavior of CLZ encapsulated in the mesoporous materials (SBA-15, MCM-41, and Al-MCM-41) and in the zeolites (NaX, NaY, and HY).

The mesoporous-based composites show an average ratio of the peak intensities I_700_/I_600_ close to 1 (Appendix A), suggesting almost equal contribution from the CLZ populations giving rise to the two bands. It should be noted that close inspection of the ratio of the two bands for the individual particles shows a broad distribution of the corresponding values. This behavior suggests a heterogeneous loading of CLZ in the channels of the different individual mesoporous particles, which results in formation of different species in the larger channels of these materials. The average value of the I_700_/I_570_ ratio for the zeolite composites is significantly lower (Appendix A), suggesting weaker intermolecular interactions between the encapsulated CLZ molecules and that the CLZ is mostly present as a monomer-like form. Additionally, the values of I_700_/I_570_ for the individual crystals are more narrowly distributed, suggesting more homogeneous loading of CLZ across the different zeolite crystals. 

Next, we deconvoluted the normalized emission spectra of the silica-based materials loaded with CLZ. The deconvolution yields four bands centred at ~535 nm, 572 nm, 605 nm, and ~700 nm (Figure 6, Appendix A). CLZ@SBA-15, CLZ@MCM-41, and CLZ@Al-MCM-41 show similar behavior with the band at 605 nm having the highest relative contribution (50–54%), followed by the band at 700 nm (33–36%). The bands at 535 nm and 572 nm have much lower contribution (4–7% and 6–10%, respectively). For the zeolite composites, the band at 605 nm remains the one with the highest relative contribution (54%, 48%, and 61% for CLZ@NaX, CLZ@NaY, and CLZ@HY, respectively). However, the contribution of the band at 700 nm decreases significantly for all the zeolites, (14%, 23%, and 27% for CLZ@NaX, CLZ@NaY, and CLZ@HY, respectively). The band at 572 nm has a comparable contribution (7–8%) with the one observed for the mesoporous composites, with the notable exception being CLZ@NaY (16%). Finally, the contribution of the band at 535 nm increases slightly for CLZ@NaX and CLZ@NaY (16 and 21%, respectively) and is 6% for CLZ@HY. 

We assign the bands at 535 and 700 nm to different aggregate populations, while the bands at 572 nm and 605 nm to the neutral form of CLZ and protonated CLZ^+^, respectively. In the study of CLZ@MSPs, the low amount of silanol groups (1% and 5% for the hydrophobic particles and the more hydrophilic ones, respectively), favors the presence of the neutral CLZ as a result of the reduced H-bonding ability of these materials [23]. In the composites in the current study, CLZ^+^ is the dominant species as evidenced also by the steady-state absorption and emission spectra. The emission band at 700 nm reported also for CLZ@MSPs [23], suggests the presence of J-type aggregates [39,42,50,51,52,53]. The contribution of the band at 700 nm is higher for the mesoporous-based composites (CLZ@SBA-15 (36%), CLZ@MCM-41 (33%), and CLZ@Al-MCM-41 (33%)), compared to the zeolite ones (CLZ@NaX (14%), CLZ@NaY (23%), and CLZ@HY (23%)), which can be related to the higher loading for the former group. The bigger channels (25–60 Å) and higher loading (80–100%) for the mesoporous materials compared to the zeolites (7.4–13 Å; 40–51%) can facilitate the formation of head-to-tail interactions that lead to loose J-type aggregates. It should be noted that we did not observe the 700 nm band in the steady-state emission spectra, which can be a consequence of a solvation process that reduces the aggregation or due to the effect of ensemble averaging in the case of the steady-state emission spectra measured in suspension. 

Finally, the band at 535 nm, which contributes no more than 7%, except for CLZ@NaX and CLZ@NaY, arises from a population of H-type aggregates [42,52,53]. For the latter materials, the contributions of the 535 nm band to the emission spectra of CLZ are 16% and 21%, respectively. Due to the size of the zeolite channels, two effects are expected: (a) hindering head-to-tail interactions and (b) promoting a tighter face-to-face interactions, typical of H-type aggregates. 

#### 2.5.2. TCSPC Decays of Solid Composites

To study further the interactions of CLZ with the silica-based materials, we excited the solid composites at 470 nm and collected the emission decays using a 510–570 nm bandpass filter (Filter I) and a 700 nm long-pass filter (Filter II). The emission decays for representative crystals are shown in Figure 2, Figure 3, Figure 4 and Figure 5 and Appendix A (panels C and D for Filter I and Filter II, respectively). The values of the averaged time constants and their relative pre-exponential factors obtained from a multiexponential fit of the experimental data are shown in Table 3 and Table 4. The data for each single crystal are reported in the Appendix A.

The emission decays collected using Filter I and Filter II show multiexponential behavior for all the composites. The fits gave three time components with average value of τ_1_ = 0.1 ns, τ_2_ = 0.4–0.7 ns and τ_3_ = 2.6–3.6 ns for Filter I, and τ_1_ = 0.1 ns, τ_2_ = 0.4–0.9 ns and τ_3_ = 1.3–2.2 ns for the decays collected using Filter II. Several composites of SBA-15, MCM-41, and Al-MCM-41 did not show the 0.1 ns time component, giving rise to bi-exponential decays with the values of τ_2_ and τ_3_ comparable to the other samples (See Appendix A). The emission intensity of the empty hosts was low (10–30 photon counts) in comparison to the CLZ composites (>1000 photon counts) when collected under the same conditions. Therefore, the contribution of the empty materials to the emission decays is negligible. We assign τ_1_ to a strong intermolecular CLZ–CLZ interactions leading to formation of aggregates. This component, which is limited by the time resolution of the system, is present in the decays collected using both filters. Its relative contribution depends on the type of silica-based host.

In the mesoporous composites, τ_1_ is predominantly present in the decays at the blue side of the emission spectrum (Filter I), while at the red side (Filter II), it is found in only few of the studied particles. On the contrary, τ_1_ is consistently present in almost all of the CLZ@zeolite composites emission decays independently on the interrogated spectral range. This behavior suggests that τ_1_ corresponds to different types of aggregates and due to the limited time resolution, it is not possible to obtain its correct values. To confirm the correct assignment of this component to strong CLZ–CLZ interactions, we collected emission decays of single crystals of CLZ@NaX and CLZ@MCM41 at significantly lower CLZ loading (initial CLZ concentration during the loading procedure was between 10^−9^–10^−7^ M). For both composites, the emission decays become bi-exponential (τ_2_ = 0.9–1.6 ns (73%) and τ_3_ = 5.2–5.7 ns (27%) for Filter I, and τ_2_ = 0.6–0.9 ns (85–100%) and τ_3_ = 2–4 ns (0–15%) for Filter II) with the shortest component of 0.1 ns absent. Similar concentration dependent emission dynamics was reported for CLZ loaded in MSPs [23]. We observed time values in the sub-nanosecond range at high CLZ loading, which became longer when we decreased CLZ concentration in the pores. In the hydrophobic particles, the sub- nanosecond component remained due to the higher hydrophobic CLZ–MSPs interaction that promotes the aggregation, while for the hydrophilic particles both time values increased above 1 ns, indicating weaker CLZ–MSPs interactions and higher amount of monomer dispersed in the pores. 

The second component, τ_2_, arises from weak CLZ–CLZ interactions. For the emission decays of the mesoporous-based composites, τ_2_ = 0.6–0.7 ns when using Filter I and τ_2_ = 0.8–0.9 ns at the red side of the spectrum (Filter II). For the zeolite-based composites, these values are ~0.4–0.6 ns, almost independently of the studied spectral range. The spread of the values (measured as the standard deviation, σ) of τ_2_ across the studied single crystals also depends on the type of the host. For CLZ@MCM41, CLZ@Al-MCM41, and CLZ@SBA-15, σ is between 0.09 and 0.19, while for the CLZ@zeolite composites, it is in the range of 0.02–0.09 (Table 3 and Table 4) with the differences being more significant for the emission decays collected using Filter II. As a result, the emission decays of the zeolite-based composites are almost identical (Figure 4 and Figure 5 and Appendix A, panels C and D), while the ones corresponding to the CLZ@mesoporous hosts are more diverse across the particles of the same material (Figure 2 and Figure 3, and Appendix A, panels C and D). This behavior is in agreement with the spectral one and is most probably associated with the structural properties of the different host families. The smaller channels of the zeolites allow for limited number of molecules and thus molecular orientations to be accommodated. On the contrary, although the loading is much higher for the mesoporous materials (80–100%, Table 1), the larger channel size allows for larger variety of molecular orientations and interactions. Additionally, the zeolite framework is crystalline, while the mesoporous materials are amorphous, which can further influence the heterogeneity in the observed spectral behavior for the latter family.

Finally, the average values of τ_3_ are comparable for all the hosts. Due to the lower relative contribution of this component, a reliable standard deviation could not be estimated. At the blue side of the emission spectra, τ_3_ = 2.6–3.8 ns (Filter I), with an average relative amplitude, a_3_ = 6–8% for the mesoporous family of composites and a_3_ = 1–4% for the zeolite-based ones. At the red side of the spectra (Filter II), τ_3_ = 1.3–2.1 ns and a_3_ = 8–15% and 2–5% for CLZ@mesoporous and CLZ@zeolite hosts, respectively. We assign τ_3_ to the emission of monomer-like populations. At high CLZ loading, the drug is present in the channels of both the mesoporous and zeolite materials predominantly in the aggregated form and as a result, the relative contribution from the monomer-like population to the overall emission decays is low. The steady state ensemble average and the single particle emission spectra both suggest that the value of τ_3_ obtained using Filter I corresponds mostly to the neutral form of CLZ (565 nm), while τ_3_ at the red side of the spectra (Filter II) is assigned to the protonated form. This is in agreement with the dependence of the average relative contribution of τ_3_. In the zeolite-based composites, it does not vary between the two interrogated spectral regions, while for the mesoporous composites, it has higher contribution to the decays collected using Filter II. This behavior can be explained in terms of the larger channels of the mesoporous materials that allow for higher concentration of the monomer-like population. The values of τ_3_ show larger dispersion, which is most probably associated with its low relative contribution to the overall emission decays. Upon decreasing the CLZ concentration (10^−7^–10^−9^ M), the relative contribution of τ_3_ increases to 25–30% for CLZ@MCM41 and to 35–45% for CLZ@NaX (Appendix A and Appendix A). This increase is concomitant with an increase in its value. This trend is indicative of a significant decrease in the CLZ intermolecular interaction and an increase in the monomer-like population.

## 3. Materials and Methods 

Clofazimine (CLZ; >98% pure), the mesoporous materials (SBA-15, MCM-41, Al-MCM-41), and the zeolites (NaX, NaY, HY) were purchased from Sigma-Aldrich (Madrid, Spain), and used as received. Dichloromethane (DCM, spectroscopic grade 99.8%) was purchased from Scharlab (Barcelona, Spain). The CLZ@silica materials composites were prepared by adding 50 mg of a dried (3 h at 550 °C) silica material into 10 mL of a DCM solution of CLZ (4.3 × 10^−5^ M). The obtained suspensions were stirred at room temperature overnight, centrifuged, and rinsed four times using pure DCM to remove the excess of weakly bound CLZ molecules. Due to the low CLZ concentration no reliable adsorption studies could be performed. The solids were dried under vacuum at room temperature (293 K). For each composite the maximum % of CLZ loaded was calculated from the absorption intensity of the supernatant measured at 450 nm during the washing procedure.

Steady-state UV-visible absorption and emission spectra were measured by means of JASCO V-670 (JASCO, Pfungstadt, Germany) and Fluoromax-4 (Jobin-Yvon, Paris, France) spectrophotometers, respectively, using the solid composites dispersed in DCM (1 mg/mL) after the washing procedure.

Fluorescence lifetime images were taken using an inverted-type scanning confocal fluorescence microscope MicroTime-200 (Picoquant, Berlin, Germany), with a 60× NA1.2 Olympus water immersion objective, and a 2D piezo scanner (Physik Instrumente, Karlsruhe, Germany). A pulsed diode laser (470 nm with a pulse width of ~40 ps and a laser power of ~20 mW, 10 MHz) was used as the excitation (~0.5 mW at the sample), presenting an instrument response function (IRF) of ~120 ps. A dichroic mirror (AHF, Z375RDC), a long-pass filter (AHF, HQ530lp), a 50 µm pinhole, a band-pass filter, and an avalanche photodiode detector (MPD, PDM series) were used to collect the emission from the solid samples. Exponential fits of the fluorescence decays were performed via iterative least-squares deconvolution using the SymphoTime software (Picoquant, Berlin, Germany). The single particle emission spectra were recorded using a spectrograph (Andor SR 303i-B) equipped with a 1600 × 200 pixel EMCCD detector Andor Newton DU-970N-BV (Oxford Instruments, Belfast, Northern Ireland) coupled to the MicroTime-200 system. All the experiments were performed at room temperature (293 K).

## 4. Conclusions

In this work, we studied the effect of the structural properties of silica-based hosts on the interactions and dynamics of encapsulated CLZ. In all the materials we observed the protonation of CLZ, its neutral form and aggregates formation. This results in a rich dynamical behavior reflected in a multiexponential emission decays with lifetime components associated with aggregates (0.1 ns), weakly interacting aggregates (0.4–0.9 ns), and neutral and protonated CLZ (1.3–3.8 ns). The relative contribution of these populations varies across the different hosts and is associated with the acidity (Lewis or Brønsted), presence of Na^+^ counterion, and the size of the pores. We observed higher contributions from J-type aggregates and monomer-like species for the mesoporous composites, while for the zeolite-based ones, the predominant contribution arises from H-type aggregates. 

Additionally, we conclude that the increased affinity of CLZ for the hydrophobic pores of MCM-41, Al-MCM-41, SBA-15, along with the presence of silanol and OH groups in the framework favors the presence of monomer-like populations. However, the large pore size allows the stacking of the CLZ molecules in J-type aggregates. On the other hand, the smaller spaces of the zeolite pores favor stronger CLZ–CLZ interactions, causing formation of H-type aggregates, while the Na^+^ screening effect observed in NaX and NaY reduces the strength of the interaction with the zeolites framework. FLIM experiments on single particles reveal that while CLZ is loaded more efficiently in the mesoporous materials, the dynamic behavior across different particles is more heterogeneous, and is associated with different relative contribution of the formed species. This is in contrast with the lower CLZ loading and more homogeneous dynamics in the zeolite-based composites. The observed heterogeneous behavior across the different mesoporous material particles might be associated with their amorphous nature as compared to the crystalline framework of the zeolites, where the behavior is more homogenous.

These results demonstrate the sensitivity of CLZ to the nature of the silica-based host materials and provide characterization of the interaction and conformation of the encapsulated drug. These results will help in the tailoring process of silica-based materials for potential drug-delivery applications.

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
