# Peer review of "Single Crystal FLIM Characterization of Clofazimine Loaded in Silica-Based Mesoporous Materials and Zeolites"

_ijms, 2019, doi:10.3390/ijms20122859_

Round 1
Reviewer 1 Report
The authors describe a study of the antibiotic Clofazimine (CLZ) dissolved in dichloromethane, CCl2H2, and then adsorbed into the mesoporous silicates MCM-41, Al MCM-41, SBA-15, and the zeolites NH4Y (HY?), NaY and NaX. The authors use diffuse transmission UV-vis and fluorescence emission spectroscopy to assess the configuration of the hydrophobic CLZ when confined in the host. The authors describe the absorbance and charge transfer processes of CLZ in pure CCl2H2 solvent, then relate the overall red-shift they observe for the adsorbed antibiotic to the formation of, variously, protonated forms of the isolated CLZ, and aggregates of the same in the silicate hosts. The dynamics of single-photon fluorescence decays were also measured at picosecond resolution. I find the manuscript to be generally well-written, results are interesting and well explained, and conclusions defensible. However, I have some concerns, especially regarding the authors apparent emphasis on geometric confinement effects as the primary cause of differences in DCM spectroscopic behavior. My concerns are discussed below; once these are addressed the manuscript may be published in substantially the current form.
· Line 36: change to ‘antibiotic used mainly to treat leprosy.’
· Line 40-41: is ‘solubility’ the right word? How, thermodynamically, is solubility affected by encapsulation? Do the authors mean ‘bioavailability’ instead of solubility?
· Line 46-47: sentence should read ‘ …present high pore volume, large surface area, and tunable pore diameter that translates to high loading capacity.’ Tunable pore volume is an attribute designed into the material, it is not an outcome translated from high pore volume and large surface area.
· Line 65: Change SBA15 to SBA-15.
· Line 109: I am not clear on what the authors mean by partial encapsulation. Is the CLZ not fully absorbed into the zeolite? Seems like the CLZ would fit quite nicely into the supercages of the X or Y (FAU) structure.
· Line 115-121: The author’s use of NH4Y to designate the material in question is inappropriate, and the general description of the material is somewhat misleading. Their zeolite was calcined at 550° C for 3 hours (Line 413). As they state, this is sufficient to drive the of ammonia (not ammonium), this leaves behind a proton to charge compensation the negative framework. This is a Brönstead acid, which will happily protonate CLZ. When the framework change is compensated by another cation (Na+) is best thought of as a Lewis acid. This is a change in the type of acidity, not an increase. The framework charge will be satisfied, whether by a proton or some other cation, and the sentence “The reduced amount of the counterion upon heating the zeolite, decreases the screening effect and allows better interaction and stabilization of CLZ with the silica…”, does not make sense to me. Shouldn’t the interaction with the framework be understood in terms of the character of the acidity and charge carrying cations.
· Line 134: the ‘presence of Na+ is greatly reduced by the use of NH4+ as counterion’. As noted above, NH4+ is likely not present. The cation will be Na+ or H+. My feeling is the different absorption and emission spectra of the HY should be understood in terms of the Brönstead / Lewis character of the materials.
· Line 135: Omit the word “Finally” and begin a new paragraph. The implied connection between the different materials is misleading. Mesoporous silicas are amorphous and largely framework neutral, very different than the crystalline Al-Si frameworks of the zeolites. This is a concern throughout the manuscript, where the immediate comparison of mesoporous materials to the zeolites suggests they should be expected to be more similar than they are. We expect them to be very different.
· Line 139: The authors give BET surface area data for the silica materials; however, discuss no relationship between surface area and CLZ sorption behavior. For zeolites, the BET analysis is widely considered to be a problematic and therefore meaningless characterization method. Total pore volume or total accessible pore volume should be included in the table, this would be a better and more useful characterization of porosity in the materials studied. How does total pore volume scale with CLZ loading?
· Line 205: sentence construction is confusing, ’even though’ suggests because the size of the channel is large enough, therefore one expects the CLZ-CLZ interaction to be weak. I think the authors mean because the pores in SBA-15 are large, this facilitates the formation of aggregates’.
· Line 212-214: Is the reasoning that ‘the smaller pores of MCM-41 … can provide multiple interaction sites for CLZ… ‘ the best? The SBA-15 has large pores, but the surface is very folded and microporous, these convolutions can result is a very higher apparent silanol density, quite unlike MCM-41. Elsewhere in the manuscript, can the apparently unique behavior of SBA-15 be attributed to the presence of these micropores?.
Author Response
Point 1: Line 36: change to ‘antibiotic used mainly to treat leprosy.’
Response 1: Changed as suggested (line 36, page 1).
Point 2: Line 40-41: is ‘solubility’ the right word? How, thermodynamically, is solubility affected by encapsulation? Do the authors mean ‘bioavailability’ instead of solubility?
Response 2: Clofazimine is nearly insoluble in water solutions at neutral pH and it rapidly precipitates in such conditions, meaning that it is not adsorbed well in the human body. Hydrophilic nanocarriers can encapsulate CLZ and maintain it in solution preventing the precipitation processes (references 5, 9-12). Additionally, in case of mesoporous materials, the nanoconfinement effect maintains CLZ in its amorphous form, which is more soluble in aqueous solutions than other crystalline CLZ forms after its release from the nanocarrier. Because of the higher solubility the antibiotic is absorbed better in the GI tract, with higher concentration and exerts a more efficient antibacterial activity (lines 41-46, pages 1-2).
Point 3: Line 46-47: sentence should read ‘ …present high pore volume, large surface area, and tunable pore diameter that translates to high loading capacity.’ Tunable pore volume is an attribute designed into the material, it is not an outcome translated from high pore volume and large surface area.
Response 3: Changed as suggested by the reviewer (line 51, page 2).
Point 4: Line 65: Change SBA15 to SBA-15.
Response 4: Changed as suggested by the reviewer (line 70, page 2).
Point 5: Line 109: I am not clear on what the authors mean by partial encapsulation. Is the CLZ not fully absorbed into the zeolite? Seems like the CLZ would fit quite nicely into the supercages of the X or Y (FAU) structure.
Response 5: The size of the supercages of zeolites NaX and NaY is 1.3 nm but the windows connecting them are only 0.8 nm wide, whereas CLZ is 0.75-0.92 nm wide, as represented in Scheme I, which could lead to partial encapsulation of CLZ within the pores of the zeolites. We added the information about the 0.8 nm windows and the dimensions of CLZ in the text to clarify this assumption and added a reference to Scheme I to provide further visualization (lines 115-117 and lines 142-144, page 4).
Point 6: Line 115-121: The author’s use of NH4Y to designate the material in question is inappropriate, and the general description of the material is somewhat misleading. Their zeolite was calcined at 550° C for 3 hours (Line 413). As they state, this is sufficient to drive the ammonia (not ammonium), this leaves behind a proton to charge compensation the negative framework. This is a Brönstead acid, which will happily protonate CLZ. When the framework change is compensated by another cation (Na+) is best thought of as a Lewis acid. This is a change in the type of acidity, not an increase. The framework charge will be satisfied, whether by a proton or some other cation, and the sentence “The reduced amount of the counterion upon heating the zeolite, decreases the screening effect and allows better interaction and stabilization of CLZ with the silica…”, does not make sense to me. Shouldn’t the interaction with the framework be understood in terms of the character of the acidity and charge carrying cations.
Response 6: The Y zeolites containing NH4+ as counterion that have been heated at 550° C are now named HY throughout the text and figures, as the previous abbreviations was misleading. The presence of H+ as counterion after the calcination process has been clarified. The different photobehaviour of CLZ within NaX NaY and HY zeolites has been discussed in terms of different charges affecting the strength of the screening effect and Bronsted and Lewis acidity. We suggest that a Bronsted acidity tends to promote the protonation of CLZ at a higher degree, causing a longer red shift in the absorption spectra, as commented by the reviewer (lines 123-129, page 4).
Point 7: Line 134: the ‘presence of Na+ is greatly reduced by the use of NH4+ as counterion’. As noted above, NH4+ is likely not present. The cation will be Na+ or H+. My feeling is the different absorption and emission spectra of the HY should be understood in terms of the Brönstead / Lewis character of the materials.
Response 7: We agree and changed the text referring to HY and to the presence of the H+ counterion. We consider that the presence of a counterion Na+/H+ and small pores sizes impedes the loading for the zeolites, compared to ordered mesoporous materials that present large pores and no counterions (lines 136-146, page 4). The Lewis and Bronsted acidity effect on the photophysical behavior of CLZ is discussed in the following paragraphs.
Point 8: Line 135: Omit the word “Finally” and begin a new paragraph. The implied connection between the different materials is misleading. Mesoporous silicas are amorphous and largely framework neutral, very different than the crystalline Al-Si frameworks of the zeolites. This is a concern throughout the manuscript, where the immediate comparison of mesoporous materials to the zeolites suggests they should be expected to be more similar than they are. We expect them to be very different.
Response 8: We agree with the reviewer, the word “finally” has been removed and a new paragraph has been started (lines 152, page 4). We also agree that the crystallinity of the materials might play a role in the heterogeneous behavior across the different mesoporous particles (amorphous solid) as compared to the more homogeneous one in the crystalline zeolites. We have clarified this part in the conclusion (lines 486- 489, page 14).
Point 9: Line 139: The authors give BET surface area data for the silica materials; however, discuss no relationship between surface area and CLZ sorption behavior. For zeolites, the BET analysis is widely considered to be a problematic and therefore meaningless characterization method. Total pore volume or total accessible pore volume should be included in the table, this would be a better and more useful characterization of porosity in the materials studied. How does total pore volume scale with CLZ loading?
Response 9: The BET surface has been removed from Table 1. The pores volume of the materials is now shown in Table 1 and discussed in paragraph 2.2 (line 135 onward, page 4). CLZ loading is taken in consideration discussing the different loading capacity of the composites (line 147, page 4).
Point 10: Line 205: sentence construction is confusing, ’even though’ suggests because the size of the channel is large enough, therefore one expects the CLZ-CLZ interaction to be weak. I think the authors mean because the pores in SBA-15 are large, this facilitates the formation of aggregates’.
Response 10: The phrase has been changed relating the presence of aggregates to the large pores that can facilitate their formation, which also suggest the presence of strong CLZ-CLZ interactions. (line 205, page 6).
Point 11: Line 212-214: Is the reasoning that ‘the smaller pores of MCM-41 … can provide multiple interaction sites for CLZ… ‘ the best? The SBA-15 has large pores, but the surface is very folded and microporous, these convolutions can result is a very higher apparent silanol density, quite unlike MCM-41. Elsewhere in the manuscript, can the apparently unique behavior of SBA-15 be attributed to the presence of these micropores?.
Response 11: A comment on the roughness of SBA-15 surface, compared to MCM-41, has been added (lines 205-208, page 6). The lack of defects in MCM-41 and Al-MCM-41 and the smaller pores are addressed as the reason of reduced amount of CLZ aggregates, compared to SBA-15 (lines 247-249, page 7).
Reviewer 2 Report
In the submitted manuscript deposition of clofazimine (CLZ) on the surface of various microporous (X and Y zeolites) and mesoporous (MCM-41 and SBA-15 silicas) materials was studied. This approach seems to be important due to development of new drug delivery platforms. The porous silica-based materials can be useful carriers for such applications. The manuscript presents the results collected for the next step of research, which was previously reported for two mesoporous silicas (with average pore size diameters of 15-16 nm) synthesized by the sol-gel technique process [Phys. Chem. Chem. Phys., 2018, 20, 11899-11911]. The submission could be considered for publication, but some crucial revision is needed:
1. The mechanism of interactions between CLZ and the support surface, which is discussed in the manuscript, could be more reliable after adding FTIR studies to the revised version of this work. The DRIFT (or PAS) results could show the state of CLZ and support after deposition. The changes in the spectra above 3000 cm-1 would be helpful to verify a participation of silanols in bonding of CLZ.
2. In many points the efficiency of CLZ deposition is related to hydrophilicity and/or acidity of the selected supports (for example the text between line 165 and 178). However, these properties have not studied. For example, what is difference in acidity/hydrophilicity between MCM-41 and Al-MCM-41? I expect that after the introduction of Al these parameters should increase. Anyway, the authors should determine acidity experimentally for all studied materials.
3. In the case of the microporous zeolite supports, deposition of CLZ occurred probably only on the external surface of particles due to too narrow channels limiting diffusion of organic molecules. To exclude this supposition, the CLZ loadings in the modified samples should be determined by TGA combined with chemical analysis and the obtained results should be discussed in relation to a possible monolayer of CLZ formed on the external surface (calculated based on the N2 adsorption isotherms).
4. Please, verify in line 416 if adsorption of dye was really studied?
Author Response
Point 1: The mechanism of interactions between CLZ and the support surface, which is discussed in the manuscript, could be more reliable after adding FTIR studies to the revised version of this work. The DRIFT (or PAS) results could show the state of CLZ and support after deposition. The changes in the spectra above 3000 cm-1 would be helpful to verify a participation of silanols in bonding of CLZ.
Response 1: Unfortunately the low concentration (<4 % w/w) of CLZ within the silica materials used for the spectroscopic study prevents from obtaining meaningful information from the FTIR analysis, as can be observed in the Supplementary Material (line 19, page 1) Figure S2 and at the end of the comments. We have added a comment on this on line 108, page 3.
Point 2: In many points the efficiency of CLZ deposition is related to hydrophilicity and/or acidity of the selected supports (for example the text between line 165 and 178). However, these properties have not studied. For example, what is difference in acidity/hydrophilicity between MCM-41 and Al-MCM-41? I expect that after the introduction of Al these parameters should increase. Anyway, the authors should determine acidity experimentally for all studied materials.
Response 2: The discussion regarding the effect of acidity and Si/Al substitution is presented now in terms of Lewis and Bronsted acidity, depending on the amount of Al and the presence of the counterions (for the zeolites) rather than absolute amount of acidity as it would not be possible to provide measured acidity at this present time. (line 189-197, page 5)
Point 3: In the case of the microporous zeolite supports, deposition of CLZ occurred probably only on the external surface of particles due to too narrow channels limiting diffusion of organic molecules. To exclude this supposition, the CLZ loadings in the modified samples should be determined by TGA combined with chemical analysis and the obtained results should be discussed in relation to a possible monolayer of CLZ formed on the external surface (calculated based on the N2 adsorption isotherms).
Response 3: We agree that CLZ may not fit properly into the cavities of the zeolites and its diffusion deep within their framework could be hampered. However, all the loaded composites have been rinsed four times with dichloromethane (lines 447-449, page 13) in order to remove the loose CLZ molecules. The remaining molecules are the ones interacting more strongly with the framework and that are at least partially encapsulated in the pores. The low concentration of CLZ loaded into the zeolites (1.2-1.8 %w/w, Scheme I) would not allow for reliable TGA results as in the case of FTIR analysis.
Point 4: Please, verify in line 416 if adsorption of dye was really studied?
We have indicated in the text that due to the low CLZ concentration no additional adsorption studies have been performed.
Round 2
Reviewer 2 Report
CLZ is not a dye, therefore the mistake in line 452 should be corrected.
Author Response
Point 1: CLZ is not a dye, therefore the mistake in line 452 should be corrected.
Response 1: We have corrected the text according to the reviewer´s suggestion